# The Stability of Analytes of Ionized Magnesium Concentration and Its Reference Range in Healthy Volunteers

**DOI:** 10.3390/biomedicines11092539

**Published:** 2023-09-15

**Authors:** Juhaina Salim Al-Maqbali, Abdullah M. Al Alawi, Zubaida Al-Falahi, Henrik Falhammar, Ibrahim Al-Zakwani, Mohammed Al Za’abi

**Affiliations:** 1Department of Pharmacology and Clinical Pharmacy, College of Medicine and Health Science, Sultan Qaboos University, Muscat 123, Oman; juhaina@squ.edu.om (J.S.A.-M.); azakwani@squ.edu.om (I.A.-Z.); zaabi@squ.edu.om (M.A.Z.); 2Department of Pharmacy, Sultan Qaboos University Hospital, Muscat 123, Oman; 3Department of Medicine, Sultan Qaboos University Hospital, Muscat 123, Oman; z.alfalahi@squ.edu.om; 4Internal Medicine Residency Training Program, Oman Medical Specialty Board, Muscat 130, Oman; 5Department of Endocrinology, Karolinska University Hospital, 17176 Stockholm, Sweden; henrik.falhammar@ki.se; 6Department of Molecular Medicine and Surgery, Karolinska Institute, 17176 Stockholm, Sweden

**Keywords:** reference range, stability, ionized magnesium, whole blood, direct ion selective electrode technique

## Abstract

This study aimed to determine the stability of refrigerated analytes of iMg concentration at different time intervals and to establish iMg reference range in a cohort of healthy Omani volunteers (≥18 years). The concentrations of iMg were measured using the direct ion-selective electrode technique. Pearson’s and Lin’s concordance correlation coefficients along with the Bland–Altman plot were used to assess the levels of agreement between iMg concentrations of fresh and refrigerated blood samples at different time intervals. The study included 167 volunteers (51% females) with a median age of 21 (range: 20–25) years. The median, 2.5th, and 97.5th percentiles for fresh iMg reference ranges were 0.55, 0.47, and 0.68 mmol/L, respectively. The overall agreement between the fresh and refrigerated iMg concentrations was poor (*rho-c* = 0.51; *p* < 0.001). However, according to Altman’s definition, iMg concentrations of the refrigerated samples for a period of ≤1 h had an excellent correlation with the fresh iMg concentrations (Lin’s *rho-c* = 0.80), with a small average bias difference of 0.009 (95%CI; −0.025–0.043). A cut-off refrigeration period within ≤1 h at 2–8 °C can be considered an alternate time frame for the gold standard measurement (fresh or within 0.5 h).

## 1. Introduction

Magnesium (Mg) serves as a cofactor for over 300 enzymes within the human body, regulating a variety of vital functions such as glycemic control, myocardial and muscle contraction, and blood pressure [1,2]. Mg is absorbed in the small intestine, maintained by renal re-absorption, and excreted through the kidneys [3,4]. It is reported that the total magnesium (tMg) is maintained at a concentration of 0.7–1.0 mmol/L [1,5].

Around 99% of the total body Mg is found intracellularly, while the extracellular Mg is either bound to albumin (25%), complexed with anions (8%), or free as bioactive ionized form (65–70%) [4,6]. Therefore, total body Mg is a poor indicator of Mg status [3], and measuring ionized magnesium (iMg), which is biologically the active form, has been found to be a more sensitive and reliable marker [6,7,8]. Furthermore, because iMg is not bound to albumin, it is thought to be a more accurate representation of Mg concentrations in the body than tMg in critical care settings where patients’ hemodynamics are changing rapidly [1].

Reports from observational studies and clinical trials did not show a statistically significant positive correlation between tMg and iMg using Pearson’s correlation coefficients (range r = 0.50 to r = 0.77) [9,10,11]. In addition, a reduction in iMg concentrations was observed despite the presence of normal tMg concentrations [7,9,10,11,12,13,14,15,16].

Prior analytical cohorts from different populations, including Asian, European, Canadian, and American, yielded a considerable variation in reference ranges. These ranges spanned from a lower cut-off limit of 0.44 to 0.51 to a higher cut-off limit of 0.60 to 0.76 mmoL/L [11,16,17,18,19]. These observed variations in iMg reference ranges in a different population are most likely attributed to variations in dietary Mg intake, which can be influenced by differences in nations’ dietary habits [3,20,21]. Some common nutritional sources of Mg in food include nuts (almonds and cashews), bananas, broccoli, green vegetables (spinach), oatmeal, seeds (pumpkin, sesame, and sunflowers), soybeans, sweet corn, and whole grains [3]. People from the Eastern Mediterranean regions, including those of the Sultanate of Oman and other Gulf countries, showed deficiency in several minerals due to their cultural dietary habits and dietary resources [22].

The distribution of circulating Mg depends on several factors, including ionic strength, the presence of competing ions, temperature, and blood pH [10]. iMg concentration can be determined by anodic stripping voltammetry (ASV) and the direction selective electrode (ISE) techniques [6] which is the most commonly used method [6]. The selectivity properties of Mg-ISE change over time, and the stability of the sample becomes a concern. Thus, it is often necessary to perform the simultaneous analysis of calcium, pH, and proteins [23].

The reference range for iMg was determined in different settings. However, the results were inconsistent, most likely due to pathophysiological interference and variation in the analytical methodologies [23]. Furthermore, the stability of analytes obtained for determining iMg concentration has not been established [24]. Such discrepancies in measurement methods and questionable stability, iMg concentration, and its use as a guide in treatment are not common in clinical practice.

Therefore, in this study, we aimed to establish the optimal refrigeration time for the analyte stability to obtain a reliable iMg concentration as an alternative to the gold standard procedure for iMg measurements (fresh or within 0.5 h). We also aimed to determine the reference range of iMg concentration using healthy adult (≥18 years) Omani volunteers as a population cohort.

## 2. Materials and Methods

### 2.1. Study Design, Setting, and Population

The study was an analytical cohort that included healthy adult Omani volunteers (≥18 years), and was carried out from February to March 2023 at Sultan Qaboos University (SQU), Muscat, Sultanate of Oman. The study was approved by the Medical and Research Ethics Committee at the College of Medicine and Health Sciences, SQU, Muscat, Oman (MREC #2852; SQU-EC/006/2023).

SQU is a public high educational institution with a diverse community of people from all over the country (urban and rural), and its population age ranges from young students to older staff employees.

### 2.2. Study Subjects

All SQU staff and students were invited, via email, to participate in the study. Volunteers were accepted until the two-week collection period ended. The inclusion criteria, based on self-reporting, were not known to have medical comorbidities and not taking chronic medications or using Mg supplements. Participants were excluded if they were known to have any medical condition or if they were on chronic medication/s and using supplements containing Mg. A checklist of questions by the researcher was used to verify self-reported health status and medication use for all volunteers before accepting them to participate. Written informed consent was obtained from the participants before blood collection.

### 2.3. Preanalytical Blood Sampling

Following the application of a tourniquet, a nurse obtained a 2 mL venous whole blood sample from each participant. The syringes used were heparinized with 60 IU heparin (self-filling sampler by Radiometer ^®^, 2700 Bronshoj, Denmark) [9]. Samples were either immediately analyzed or refrigerated at 2–8 °C for later analysis [9].

### 2.4. Analytical Procedure for iMg Measurement

As per the manufacturer of the analyzer’s (Stat Profile Prime Plus^®^, Nova Biomedicals, Waltman, MA, USA) guidelines, the recommended “gold standard procedure” for measuring iMg entails conducting the measurement either immediately (freshly) or within 0.5 h of blood collection, while the sample at room temperature.

After each sample was freshly (immediately—time zero) taken and analyzed, the remaining blood was promptly refrigerated at 2–8 °C [9]. Thereafter, a batch of samples (each ~10) was randomly selected and removed from the refrigerator at different time intervals of 1, 2, 3, 4, 5, 10, 15, 25, and 30 h. These were then gently shaken and reanalyzed for iMg concentration. The obtained results were then compared to their corresponding results obtained using the gold standard procedure (fresh measurement).

### 2.5. iMg Detection Method

The electrolytes analyzer, Stat Profile Prime Plus^®^ (Nova Biomedicals, Waltman, MA, USA), uses the direct ISE method to measure the concentration of iMg. This method utilizes a neutral carrier-based membrane containing an ionophore that exhibits selectivity towards the size of the Mg ion [18].

### 2.6. Data Reporting

The assessment of stability was carried out according to CRESS checklist [25]. Age and gender, as part of the demographics of the cohort, were collected. The sampling, refrigerating, and analyzing times were collected. In addition to iMg concentration, the analyzer also measured the concentration of ionized calcium, serum creatinine, and blood pH.

### 2.7. Statistical Analysis

Categorical variables were presented as frequencies and percentages. Continuous abnormally distributed variables were summarized using the median and interquartile ranges (IQRs). The reference range was derived using the 2.5th and 97.5th percentiles. Wilcoxon–Mann–Whitney test was used to examine differences in fresh iMg concentrations by gender. Spearman’s rank correlation coefficient was used to test the effect of blood pH on fresh and refrigerated iMg concentrations. Pearson’s correlation coefficient and Lin’s concordance correlation coefficients were used to determine the correlation between fresh and refrigerated iMg concentrations. Bland–Altman plot was used to determine and illustrate the mean differences between the fresh and refrigerated iMg concentrations. Lin’s concordance correlation coefficient test was used to determine the concordance of the bivariate sets of observations to the “gold standard” measure after being refrigerated for ≥1 h. The null hypothesis of equal means and variances, which should not be rejected (*p* > 0.05), was tested using the Bradley–Blackwood procedure [26,27]. Altman [26] has suggested that Lin’s concordance correlation coefficients should be interpreted close to other correlation coefficients like Pearson’s, with <0.2 as poor and >0.8 as excellent. However, McBride has recommended another set of thresholds for the interpretation, with Lin’s concordance correlation coefficient values >0.99 as almost perfect, 0.95–0.99 as substantial, 0.90–0.95 as moderate, and <0.90 as poor [27]. However, for this study, Altman’s interpretation has been used because McBride’s definitions are too sensitive to the small changes in the concentrations that might be clinically valid, taking into account a small 95% CI.

The two-tailed level of significance was set at *p* < 0.05 level. Statistical analyses were conducted using STATA version 16.1 (STATA Corporation, College Station, TX, USA).

## 3. Results

A total of 212 responses from volunteers were received, of whom only 167 volunteers appeared and participated in the blood sample collection. Analyses were performed for fresh iMg concentrations for all the samples (*n* = 167), but due to low blood volume remaining in some of the samples, only 137 samples were used for refrigerated iMg concentrations. Blood pH was determined from the fresh blood samples of 87 volunteers and refrigerated blood samples from 68 participants.

As shown in Table 1, among the cohort, 51% (85/167) were females.

The median age was 21 (20–25) years. The median pH for the fresh samples was 7.37 (7.35–7.40). The median, 2.5th, and 97.5th percentiles for fresh iMg reference range was 0.55 (0.47 to 0.68) mmol/L, respectively. As shown in Figure 1, the median iMg concentration was lower in females than in males (0.54 versus 0.57 mmol/L; *p* < 0.001).

Figure 2 illustrates a significant negative correlation between blood pH and iMg concentrations (both fresh and refrigerated). Increasing pH levels were significantly associated with decreasing concentrations of both fresh (Spearman’s *rho* = −0.22, *p* = 0.04) and refrigerated iMg concentrations (Spearman’s *rho* = −0.42, *p* < 0.001) over time.

Bland–Altman plot illustrated the overall agreement between fresh and refrigerated iMg concentrations in Figure 3. The percentage difference was 4.38%, with a mean difference of 0.02 + 0.05 average being outside the limits of agreement. Lin’s concordance correlation coefficient showed a poor *rho-c* of 0.51; *p* < 0.001. Furthermore, the equal means and variances assumption using the Bradley–Blackwood procedure was also rejected (*p* < 0.001).

Table 2 shows the agreement between fresh and refrigerated iMg concentrations over the selected refrigeration times. According to Altman’s definition [26], the refrigerated samples for a period of ≤1 h resulted in refrigerated iMg concentrations of an excellent correlation with the fresh iMg concentrations (Lin’s *rho-c* = 0.80) and the equal means and variances using the Bradley–Blackwood procedure was not rejected (*p* = 0.183) with a very small average bias difference of 0.009 (95% confidence interval (CI); −0.025–0.043). Although the agreement between fresh and refrigerated iMg concentrations, ≤2 h appears to be excellent (Lin’s *rho-c* = 0.85); the equal means and variances using the Bradley–Blackwood procedure was rejected (*p* < 0.001). As the refrigerating time increases (>2 h to 30 h), the agreement between fresh and refrigerated iMg concentrations became poorer (Lin’s rho-c from 0.76 to 0.15), and the equal means and variances using the Bradley–Blackwood procedure were rejected (*p* < 0.001 for all).

## 4. Discussion

This study has established two important results. First, it identified the impact of varying refrigeration durations on the accuracy of iMg concentration measurements in blood analytes. A refrigeration window of 1 h at 2–8 °C was determined as a cut-off time frame for obtaining iMg concentration that is comparable to the recommended gold standard procedure. Second, it estimated a reference range of 0.47 to 0.68 mmol/L iMg concentration in Omani healthy volunteers.

In this study, we attempted to identify the potentially acceptable refrigeration time to obtain an accurate and comparable iMg concentration to freshly obtained measurements. Several correlation tests, including Pearson’s correlation coefficient and Lin’s concordance correlation coefficients, were used to evaluate the relationship between the fresh iMg concentration (gold standard measure) and the refrigerated iMg concentrations (test measure). Although the overall agreement between fresh and refrigerated iMg concentrations was poor (*rho-c* = 0.51), we found that according to Altman’s definition [26], the refrigerated samples for a period of ≤1 h also resulted in iMg concentrations of an excellent correlation with the fresh iMg concentrations (Lin’s *rho-c* = 0.80), with a very small average bias difference of 0.009, which is considered clinically an irrelevant small difference. Accordingly, we concluded that we could rely on ≤1hr as a cut-off time to be used during the refrigeration of blood samples in heparinized syringes before analyzing them using the ISE technique to provide valid iMg measurements, as well as the gold standard measure (fresh sample or within 0.5 h). The identified time frame window could prove to be advantageous within a clinic that experiences high levels of daily busy activities.

The established reference range in this study is consistent with the previously identified mean reference range in a systemic review by Fairley et al. (0.44–0.60 mmol/L) [19]. It also did not differ significantly from the previously identified reference ranges using the same measurement method [16,18]. Blood samples, collected in heparinized syringes from 123 healthy adults aged 20 to 67 years (73% female) from the Netherlands, established a reference range of 0.49–0.71 mmol/L [16]. Additionally, blood samples collected in lithium heparinized syringes from 125 healthy adults from the United Kingdom and stored in Sarstedt Monovette^®^ serum gel blood collection tubes, resulted in iMg reference range of 0.45–0.60 mmol/L [18]. Furthermore, a French study, using the direct ISE technique for 80 patients who visited the emergency department, established a reference range of 0.48 to 0.65 mmol/L [24].

Our established iMg reference range, was, however, inconsistent with a few other published studies. For example, Garcia et al. established a reference range of 0.51 to 0.76 mmol/L in 564 healthy volunteers from the Netherlands using Nuclear Magnetic Resonance (NMR) as a measurement technique [11]. In addition, a Japanese study in 2012 established a reference range of 0.51 to 0.63 mmol/L for 411 adults; however, the measurement method and the inclusion criteria were not mentioned in the article [17]. Compared with other methods, ISE is a potentiometric sensor that has become a validated technique for analytical characteristics of biomarkers [6,28]. They are rapid and accurate, use small sample volumes, and are a robust backbone of electrolyte measurements in the clinical laboratory for hospital use and clinical trials [29].

We identified in this study a significantly lower iMg concentration in females than in males. This finding could be explained by the observation that our participant female population had a higher pH and lower serum creatinine than their male counterparts. Mg concentration is maintained by renal re-absorption and excreted via the kidneys [3]. At the same time, blood temperature and pH are major factors affecting the intracellular and extracellular distribution of the circulating Mg, and its binding to protein increases as the pH is increased [6]. This physicochemical effect was confirmed in our cohort when refrigerated samples from 1 h to 30 h were analyzed. The iMg concentrations were found to be significantly decreased as the blood pH increased with longer refrigerating time.

This study is not without limitations. Although the study sample size is acceptable for an analytical cohort to evaluate a reference range, a larger sample would be more robust. Most of the respondents/participants were educated and were either students or in full-time employment, causing a lack of comprehensive educational and socioeconomic variation, both of which are known to influence a person’s dietary habits. Adding to that, participants’ dietary magnesium intake was not assessed. Additionally, the impact of fasting/feeding were not standardized among the participants. The variation in body mass index (BMI) and different age groups may have different average iMg concentrations; therefore, age and BMI should be considered when comparing iMg levels. Furthermore, although one measurement technique (ISE) was used, comparing other analytical methods may provide additional data for comparison, variation, and selection purposes.

## 5. Conclusions

This study established a reference range of 0.47 to 0.68 mmol/L for iMg concentration in the Omani population using the direct ISE technique for ion detection from fresh blood samples. Furthermore, based on Altman’s interpretations of Lin’s correlation coefficients, we identified a cut-off time of 1 h at 2–8 °C for refrigerating blood samples to obtain reliable iMg concentrations during busy clinical working times, as well as the recommended gold standard measure (fresh or within 0.5 h).

## Figures and Tables

**Figure 1 biomedicines-11-02539-f001:**
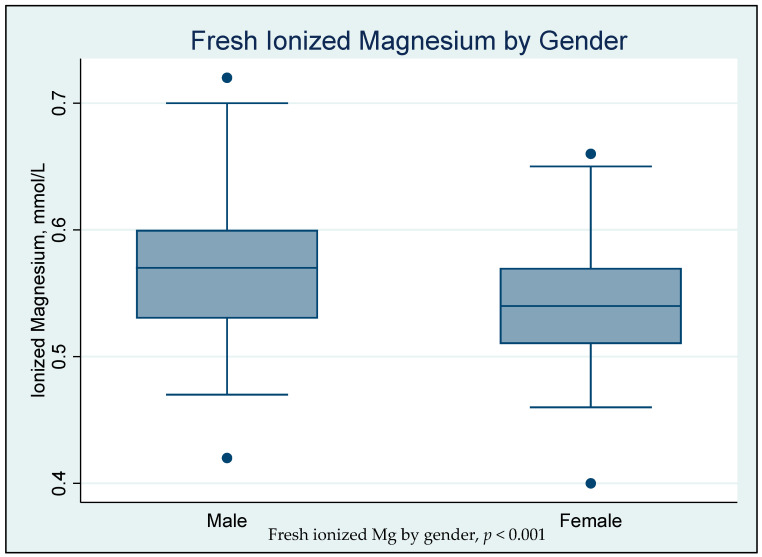
The association of fresh ionized magnesium (iMg) concentrations with gender (*n* = 167).

**Figure 2 biomedicines-11-02539-f002:**
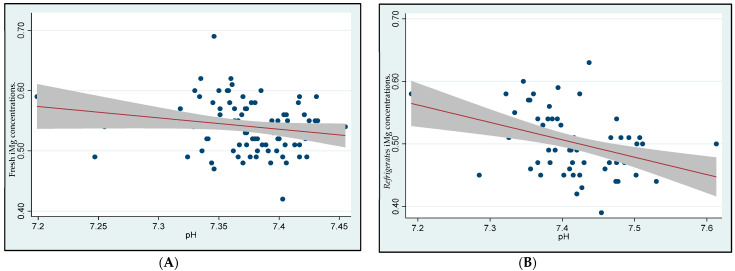
Effect of blood pH on fresh and refrigerated ionized magnesium (iMg) concentrations. (**A**) Effect of blood pH on the fresh iMg concentrations (Spearman’s *rho* = −0.22; *p* = 0.04) (*n* = 87). (**B**) Effect of blood pH on the refrigerated iMg concentrations (Spearman’s *rho* = −0.42, *p* < 0.001) (*n* = 68).

**Figure 3 biomedicines-11-02539-f003:**
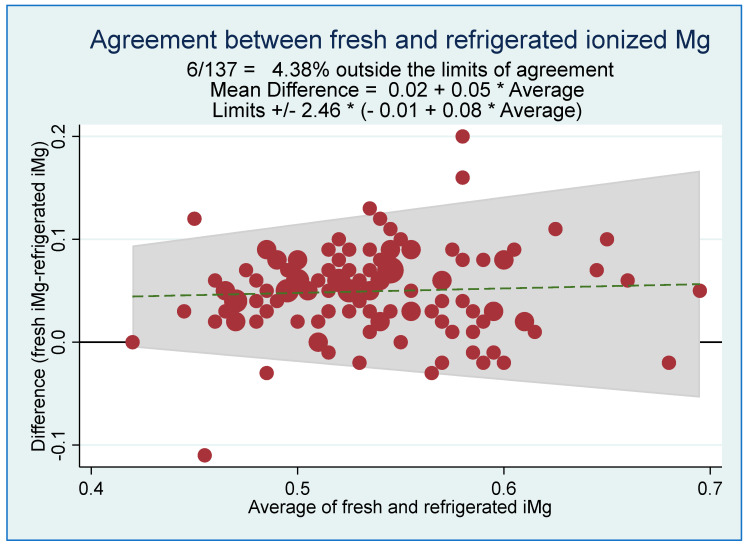
Agreement between fresh and refrigerated ionized magnesium (iMg) concentrations using Bland–Altman Polt (*n* = 137). [Person’s *rho* = 0.70; *p* < 0.001. Lin’s concordance correlation coefficient *rho-c* = 0.51; *p* < 0.001].

**Table 1 biomedicines-11-02539-t001:** Clinical characteristics of healthy Omani volunteers and the association of gender with fresh ionized magnesium, ionized calcium, and serum creatinine concentrations.

Characteristic, Median (IQR)	All (*n* = 167; 100%)	Female (*n* = 85; 51%)	Male (*n* = 82; 49%)	*p*-Value
Age, years	21 (20–25)	21 (20–23)	22 (20–30)	0.031
Blood pH^+^	7.37 (7.35–7.40)	7.38 (7.36–7.40)	7.36 (7.35–7.39)	0.028
Ionized magnesium; mmol/L *	0.55 (0.52–0.59) *	0.54 (0.51–0.57) *	0.57 (0.53–0.60) *	<0.001
Ionized calcium; mmol/L	1.28 (1.26–1.30)	1.27 (1.25–1.30)	1.28 (1.27–1.31)	0.075
Serum creatinine; mmol/L	74 (66–84)	67 (62–72)	84 (76–91)	<0.001

IQR, interquartile range (25% to 75%). Analyses were performed using the Wilcoxon-Mann-Whitney test. * 2.5th and 97.5th percentiles.

**Table 2 biomedicines-11-02539-t002:** Agreement between fresh and refrigerated ionized magnesium (iMg) concentrations (mmol/L) over studied time.

Refrigeration Time (h.)	Number of Samples (*n* = 137)	Pearson’s Correlation Coefficient (r)	Lin’s Concordance Correlation Coefficient (*rho-c*)	Bradley-Blackwood *p*-Value *	Average Difference (Bias)	95% Limits of Agreement (Bland and Altman)
≤1	14	0.84	0.80	0.183	0.009	−0.025–0.043
≤2	24	0.92	0.85	<0.001	0.017	−0.017–0.051
≤3	37	0.88	0.76	<0.001	0.026	−0.022–0.075
≤4	55	0.80	0.61	<0.001	0.038	−0.022–0.098
≤5	62	0.80	0.58	<0.001	0.041	−0.020–0.103
≤10	58	0.78	0.55	<0.001	0.044	−0.019–0.106
≤15	70	0.78	0.54	<0.001	0.043	−0.018–0.105
≤25	90	0.72	0.53	<0.001	0.040	−0.032–0.111
≤30	137	0.70	0.51	<0.001	0.045	−0.035–0.126

* The null hypothesis of equal means and variances tested using the Bradley–Blackwood procedure, should not be rejected (*p* > 0.05).

## Data Availability

Data are available on request from the corresponding author.

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
