# Peer review of "The Stability of Analytes of Ionized Magnesium Concentration and Its Reference Range in Healthy Volunteers"

_biomedicines, 2023, doi:10.3390/biomedicines11092539_

Round 1
Reviewer 1 Report
This is a generally well-prepared and useful report. I only have a few comments for the authors.
Specific comments:
1. As per the journal's guidelines, the abstract should be around 200 words and a single paragraph WITHOUT subheadings.
2. Please also include an ethical approval statement in the methods section.
3. Please label the Y axis in Figures 2A and 2B.
4. The discussion of study limitations is inadequate. When determining ionized magnesium levels in blood samples, there are several potential confounding factors that can influence the results. These confounders are variables that are not the main focus of the study but can affect the interpretation of the relationship between the measured variable (iMg) and the outcome of interest. For example, iMg levels can vary with age. Different age groups may have different average magnesium concentrations, so age should be taken into account when comparing iMg levels between individuals or groups. Even body mass index and body fat percentage can impact magnesium distribution in the body. Individuals with higher BMI might have different iMg levels.
Minor edits only.
Author Response
Reviewer 1:
Comments and Suggestions for Authors
This is a generally well-prepared and useful report. I only have a few comments for the authors.
Authors’ response: Thank you for your kind remark.
Specific comments:
- As per the journal's guidelines, the abstract should be around 200 words and a single paragraph WITHOUT subheadings.
Authors’ response: We have adjusted the abstract as per suggestion.
- Please also include an ethical approval statement in the methods section.
Authors’ response: We have added the ethical approval statement as per suggestion.
- Please label the Y axis in Figures 2A and 2B.
Authors’ response: We have added the labelling as per suggestion.
- The discussion of study limitations is inadequate. When determining ionized magnesium levels in blood samples, there are several potential confounding factors that can influence the results. These confounders are variables that are not the main focus of the study but can affect the interpretation of the relationship between the measured variable (iMg) and the outcome of interest. For example, iMg levels can vary with age. Different age groups may have different average magnesium concentrations, so age should be taken into account when comparing iMg levels between individuals or groups. Even body mass index and body fat percentage can impact magnesium distribution in the body. Individuals with higher BMI might have different iMg levels.
Authors’ response: We have now adjusted the limitations to include other potential variables/confounders that could affect the results.
Reviewer 2 Report
This analytical cohort study aimed to determine the optimal refrigeration time to obtain reliable measurements of ionized magnesium (iMg) concentration in blood samples as an alternative to the gold standard fresh measurement. The study also sought to establish a reference range for iMg concentration in healthy Omani adults. Blood samples were collected from 167 volunteers and analyzed for iMg concentration immediately and after refrigeration at 2-8°C for up to 30 hours.
The overall agreement between fresh and refrigerated iMg concentrations was poor. However, refrigerating samples for up to 1 hour provided excellent correlation with fresh measurements, with minimal bias. The established reference range for iMg concentration in the studied population was 0.47 to 0.68 mmol/L. The findings identify 1 hour as an acceptable refrigeration time to obtain valid iMg concentrations during busy clinic hours. The data also provide a population-specific reference range that can aid in interpreting iMg results for Omani patients.
Limitations include small sample size, limited population representation, self-reported eligibility criteria, lack of dietary control, limited methodology, and variables around sample handling and participant factors. Improvements would include larger diverse samples, medical record review, dietary assessment, method comparisons, more extensive stability testing, reporting of variability, standardized sampling times, and accounting for prandial status.
- Small sample size (n=167) for establishing a reference range. A larger sample would provide more robust reference values.
- Limited demographic variability in the volunteer population. Most were students/staff at one university. A more diverse population sample is needed.
- Self-reported health status and medication use may not be reliable. Medical records should be reviewed to confirm eligibility.
- Dietary magnesium intake was not assessed. This can impact magnesium levels so should be controlled for.
- The effect of sample storage temperature was not tested. Room temperature storage could have been compared to refrigeration.
- Only one measurement technique (ISE) was used. Comparing different analytical methods could improve accuracy.
- Inter-individual variability in results was not reported. Reference range studies should document this.
- Stability was only assessed up to 30 hours. Longer durations should be evaluated.
- Sample size decreased over time as blood volumes ran low. Larger volumes should be collected initially.
- Diurnal variation and impact of fasting/feeding were not considered. Sampling should occur at standardized times and account for prandial status.
Author Response
Comments and Suggestions for Authors
This analytical cohort study aimed to determine the optimal refrigeration time to obtain reliable measurements of ionized magnesium (iMg) concentration in blood samples as an alternative to the gold standard fresh measurement. The study also sought to establish a reference range for iMg concentration in healthy Omani adults. Blood samples were collected from 167 volunteers and analyzed for iMg concentration immediately and after refrigeration at 2-8°C for up to 30 hours.
The overall agreement between fresh and refrigerated iMg concentrations was poor. However, refrigerating samples for up to 1 hour provided excellent correlation with fresh measurements, with minimal bias. The established reference range for iMg concentration in the studied population was 0.47 to 0.68 mmol/L. The findings identify 1 hour as an acceptable refrigeration time to obtain valid iMg concentrations during busy clinic hours. The data also provide a population-specific reference range that can aid in interpreting iMg results for Omani patients.
Limitations include small sample size, limited population representation, self-reported eligibility criteria, lack of dietary control, limited methodology, and variables around sample handling and participant factors. Improvements would include larger diverse samples, medical record review, dietary assessment, method comparisons, more extensive stability testing, reporting of variability, standardized sampling times, and accounting for prandial status.
Authors’ response: Thank you for the summary.
- Small sample size (n=167) for establishing a reference range. A larger sample would provide more robust reference values.
Authors’ response: Although small sample size is acceptable for an analytical cohort to evaluate reference range, we agree on the importance of a larger sample size for robustness. We have now added the small sample size limitation to the manuscript as per suggestion.
- Limited demographic variability in the volunteer population. Most were students/staff at one university. A more diverse population sample is needed.
Authors’ response: The university have students and staff from all regions of Oman, which we believe represent Oman as geographic diversity, However, we agree, that the age diversity is limitation in our sample. This is now mentioned in the limitation section.
- Self-reported health status and medication use may not be reliable. Medical records should be reviewed to confirm eligibility.
Authors’ response: We included apparently healthy volunteers in our study. Although volunteers self-reported, we also did a checklist measure before accepting them to participate. We have added a statement to support this to the methods.
- Dietary magnesium intake was not assessed. This can impact magnesium levels so should be controlled for.
Authors’ response: Thank you for these important comments, we have now added this to the limitation section as per suggestion.
- The effect of sample storage temperature was not tested. Room temperature storage could have been compared to refrigeration.
Authors’ response: Although hospital room temperature is considered controlled, we have not done a comparison with refrigerated storage, as the worldwide standard practice of storing collected blood samples is the refrigeration.
- Only one measurement technique (ISE) was used. Comparing different analytical methods could improve accuracy.
Authors’ response: The aim of this study was not to compare different measurement techniques but rather the effect of refrigeration time on iMg concentration. It might, however, be possible that with different techniques temperature effect on stability is less. We have added this to the limitation.
- Inter-individual variability in results was not reported. Reference range studies should document this.
Authors’ response: We have a statement about age and BMI variations to the limitation section.
- Stability was only assessed up to 30 hours. Longer durations should be evaluated.
Authors’ response: We believe the results will not change if tested for a longer duration, since the results starts to statistically become different after 2hr and therefore, after this time it is unlikely comparable concentration will be obtained.
- Sample size decreased over time as blood volumes ran low. Larger volumes should be collected initially.
Authors’ response: We agree, however, this did not affect the validity of our statistical analysis. Such concern will be accounted for in future larger sample size studies.
- Diurnal variation and impact of fasting/feeding were not considered. Sampling should occur at standardized times and account for prandial status.
Authors’ response: We have added this to the limitation as per suggestion.